# Recent Progress on Bioinspired Antibacterial Surfaces for Biomedical Application

**DOI:** 10.3390/biomimetics7030088

**Published:** 2022-07-04

**Authors:** Xiao Yang, Wei Zhang, Xuezhi Qin, Miaomiao Cui, Yunting Guo, Ting Wang, Kaiqiang Wang, Zhenqiang Shi, Chao Zhang, Wanbo Li, Zuankai Wang

**Affiliations:** 1Department of Mechanical Engineering, City University of Hong Kong, Kowloon, Hong Kong 999077, China; xyang@hkcoche.org (X.Y.); wzhang469-c@my.cityu.edu.hk (W.Z.); xuezhi.qin@cityu.edu.hk (X.Q.); miaomicui2-c@my.cityu.edu.hk (M.C.); yuntiguo@cityu.edu.hk (Y.G.); twang259@cityu.edu.hk (T.W.); wkq19@mails.tsinghua.edu.cn (K.W.); czhan25@cityu.edu.hk (C.Z.); 2Hong Kong Centre for Cerebro-Caradiovasular Health Engineering (COCHE), Shatin, Hong Kong 999077, China; 3Centre for Nature-Inspired Engineering, City University of Hong Kong, Kowloon, Hong Kong 999077, China; 4School of Mechanical and Aerospace Engineering, Jilin University, Changchun 130022, China; 5State Key Laboratory of Tribology, Tsinghua University, Beijing 100084, China; 6CAS Key Laboratory of Separation Science for Analytical Chemistry, Dalian Institute of Chemical Physics, Chinese Academy of Sciences, Dalian 116023, China; shizhenqiang@dicp.ac.cn

**Keywords:** biomimetic, antibiofouling, surface modification, physical removal, wound dressing

## Abstract

Surface bacterial fouling has become an urgent global challenge that calls for resilient solutions. Despite the effectiveness in combating bacterial invasion, antibiotics are susceptible to causing microbial antibiotic resistance that threatens human health and compromises the medication efficacy. In nature, many organisms have evolved a myriad of surfaces with specific physicochemical properties to combat bacteria in diverse environments, providing important inspirations for implementing bioinspired approaches. This review highlights representative natural antibacterial surfaces and discusses their corresponding mechanisms, including repelling adherent bacteria through tailoring surface wettability and mechanically killing bacteria via engineering surface textures. Following this, we present the recent progress in bioinspired active and passive antibacterial strategies. Finally, the biomedical applications and the prospects of these antibacterial surfaces are discussed.

## 1. Introduction

Manifesting a typical size at least ten orders of magnitude smaller than human beings, bacteria present in various environments and are important to the human being, and ecosystem. Most bacteria are harmless to us, help our bodies digest food and absorb nutrients, and even produce multivitamins in the gut [1]. However, some diseases caused by pathogenic bacteria, such as tuberculosis, pneumonia, endocarditis, sepsis, and osteomyelitis, invade the host and cause various infectious diseases [2,3,4]. Additionally, bacteria such as methicillin-resistant *Staphylococcus aureus* and *Pseudomonas aeruginosa* are well known to trigger surgical site infections through the incision, which threatens millions of patients every year and induces the spread of antibiotic resistance all around the world [5,6]. According to the Centers for Disease Control and Prevention of the United States, antibiotic-resistant bacteria may result in at least 70,000 deaths worldwide per year. By 2050, this number will exceed 10 million [7]. 

Mitigating or even preventing bacterial infection has been a historic challenge. In ancient times, many natural agents such as herbs, honey, animal feces, and moldy bread have been widely used for treating patients with bacterial infections. Among these, the most effective and widespread agent was moldy bread, although its mechanisms were not clear at that time [8]. Meanwhile, many metals, e.g., copper and silver and their alloys, were also utilized to disinfect wounds and drinking water [9]. The discovery of penicillin was a milestone in the fight against bacterial infections, and saved thousands of wounded soldiers and civilians in wars and started the era of antibiotics and the subsequent development of new generation antibiotics. The use of systemic antibiotic therapy has been a traditional and common method for eradicating the cause of infection, yet was often unsatisfactory. For example, only a 22–37% effective rate has been reported when combating bacterial infection of medical implants such as catheters and subcutaneous sensors, because most systemic antibiotics did not reach an effective local concentration [2]. However, increasing the administrative doses of antibiotics causes cytotoxicity and side effects in the patient’s body. Another serious problem associated with the use of antibiotics is the emergence of multidrug resistance to bacterial strains, which renders current antibiotics ineffective and requires additional interventions such as more radical surgery. Therefore, ways to prevent bacterial infection and mitigate multidrug resistance simultaneously have receiving growing attention. 

Nature, however, has evolved ingenious solutions based on topological surfaces to fight bacterial infection in green and efficient manners. Typical examples of natural surfaces that exhibit antibacterial properties include the lotus leaf, wings of cicadae, wings of dragonflies, wings of planthoppers, springtail skin, shark skin, and gecko feet. Unlike antibiotic treatment, natural surfaces can physiochemically minimize bacterial infection by interfering with the surface–bacteria interaction, which fundamentally avoids the evolution of multidrug resistance [1,10,11,12,13,14,15,16,17,18]. Inspired by these elegant biological surfaces, manmade antibacterial surfaces have emerged as an efficient alternative to antibiotics for addressing bacterial challenges. In this review, we first briefly introduce the mechanism of bacterial adhesion and biofilm formation. The progress of a series of natural antibacterial surfaces is then comprehensively summarized and their antibacterial pathways discussed. Furthermore, we classify surface engineering approaches into active and passive bioinspired antibacterial surfaces, with some of their representatives being discussed in detail. Finally, recent applications of bioinspired antibacterial surfaces are illustrated, and the prospects of bioinspired antibacterial materials are proposed.

## 2. Bacterial Adhesion and Biofilm Formation 

An infection starts from the contact of an individual bacterium, during which the bacterium can actively propel itself to the surface using its flagella. When bacteria reach the surface, flagella also play an important role in adhesion by providing physical contact with surfaces, exploring local surface topography, and entering a microenvironment inaccessible to relatively large cell bodies. The interaction between flagella and surfaces could enhance adhesion, because of the inherent hydrophobicity of flagella, which allows them to adhere to hydrophobic surfaces. By contrast, the presence of flagella may also weaken adhesion as found in *Caulobacter crescentus*. Therefore, the influence of flagella on adhesion is much more complex and fully understanding it requires in-depth investigations. In addition to the flagella, some other filamentous protein extensions on the cell surface, including fimbriae, curli and pili, are also involved in nonspecific initial adhesion to abiotic surfaces [19]. For example, pili can use their specific receptors to bind to substrates through an unidentified mechanism, and most pili show no preference for substrates. Such an attachment process is sensitive to bacterial characteristics (e.g., cell development, exopolysaccharide production, metabolic activity, cell viability, cell-wall stiffness and adhesin-mediated receptor-ligand binding) and surface physicochemical properties (e.g., surface charge, surface free energy, wettability, roughness, morphology). After adhering to the surface, the cells grow, divide, and secrete exopolysaccharides to encapsulate themselves as a three-dimensional bacterial community, a so-called biofilm, in an extracellular matrix when the cell density reaches a certain level. Typical biofilms are supported by self-produced three-dimensional polymer matrix networks containing proteins, carbohydrates, nucleic acids, and other biomolecules, which can create an optimized and dynamic environment for bacterial cell growth and proliferation, protect these sessile bacterial communities from antimicrobial agents, and mediate cell-to-cell and cell-to-surface adhesion [20]. In the final stage, small pieces of cells or individual cells may be released from the biofilm and thus another infection cycle begins (Figure 1) [21,22,23].

## 3. Natural Antibacterial Surfaces

Over millions of years of evolution, nature has evolved ingenious strategies to prevent bacterial infection by breaking the transmission chain, i.e., preventing bacterial adhesion or biofilm attachment [24,25,26]. On the basis of the fundamental mechanisms, antibacterial surfaces can be classified into bacteria-repellent surfaces (e.g., marine organisms’ mucus, reptiles’ skin and plant leaves [27]) and contact-killing surfaces (e.g., insect wings) [28,29,30]. 

### 3.1. Natural Bacteria-Repellent Surface

A bacteria-repellent surface is usually achieved by introducing superhydrophobicity to remarkably lower bacterial adhesion. Superhydrophobic or the so-called self-cleaning surfaces can be widely found on plant leaves, insect cuticles, fish skin, etc., which enable these species to passively control bacterial colonization (Figure 2A). For example, a lotus leaf was the first reported to have superhydrophobicity and bacterial repellence [31]. The underpinning mechanism was the combination of low surface energy and the multiscale roughness of surface lipid structures, which allowed the surface to have a high water contact angle (*θ** 150°) and a low sliding angle (*θ*_s_ 10°), and trapped large amounts of air cushion, which significantly minimizes the surface/bacteria contact (Figure 2B). Bacterial cells colonizing such surfaces would be removed before they had a chance to form biofilms [32]. Similar phenomena have also been observed on some insect surfaces, such as planthoppers and springtails [33]. Planthoppers’ hindwings feature topographical and functional similarities to lotus leaves, thus exhibiting non-wetting behavior and low adhesion to pollutants [34,35]. Springtail skin is another kind of superhydrophobic surface consisting of a microcolumnar with a double nanoreentrant (Figure 2C) [36,37,38]. The superhydrophobic skin endowed it with an anti-adhesion property to protect springtails from bacterial attaching and infection [39]. Shark’s hydrophobic skin, leveraging flat scales or dermal denticle arrays, offers another ingenious strategy to prevent the attachment and growth of microorganisms, with additional benefits in drag reduction (Figure 2D) [40,41,42,43]. 

### 3.2. Natural Contact-Killing Surface

Unlike the bacteria-repellent strategy, many other biological surfaces violently kill the bacteria in contact with them. The contact killing effect lies in that their extremely fine structures can pierce the cell membrane due to the concentrated mechanical stress and gradually rupture the cell (Figure 2E). While varying in shape and other properties, the common feature of these natural contact-killing surfaces is their pattern in nanoscale size (50–250 nm) and two-dimensional arrangement [44]. For example, A cicada wing’s surface has uniform nanocone arrays with a height of 200 nm, a top diameter of 60 nm, a bottom diameter of 100 nm, and an interpillar space of 170 nm (Figure 2F). Unlike the lotus leaf, a cicada wing is a surface manifesting a large water contact angle of 158.8° but a high degree of bacterial adhesion. Bacteria on such a surface can be pierced through by the nanotopography [45]. Specifically, bacterial cell membranes that contact the surface patterns bear a large stretching force, accompanied by a sharp increase in the total membrane area, which collectively results in irreversible membrane rupture and bacteria death [46,47,48]. Gram-positive cells have thicker layers of peptidoglycan and are therefore generally more rigid, which may explain their increased resistance in comparison to Gram-negative cells. This is why cicadas’ wings are only effective against Gram-negative bacteria. Such functional shortcomings can be well tackled by the surface of dragonfly wings, on which both Gram-negative bacteria (*P. aeruginosa*) and Gram-positive bacteria (*S. aureus* and *Bacillus* sp.) and even endospores can be mechanically ruptured. A dragonfly wing is also covered with high aspect-ratio nanostructures that can pierce almost all bacterial membranes in contact with it (Figure 2G) [49,50]. A gecko with a unique hair structure has drawn much attention due to its superhydrophobicity and associated topographical antimicrobial effects [51]. The gecko’s skin is composed of small hairs (often called spines or microspines) a few microns in height, with an interspace of 0.2–0.7 μm (Figure 2H). Because gecko hair possesses a tip shape and size similar to the nanocones on cicadas, it can be an alternative for studying antimicrobial properties. Gecko skin has been proved to be antibacterial, with a remarkable killing effect on *Porphyromonas gingivalis*, a clinically significant bacterium [52,53]. 

**Figure 2 biomimetics-07-00088-f002:**
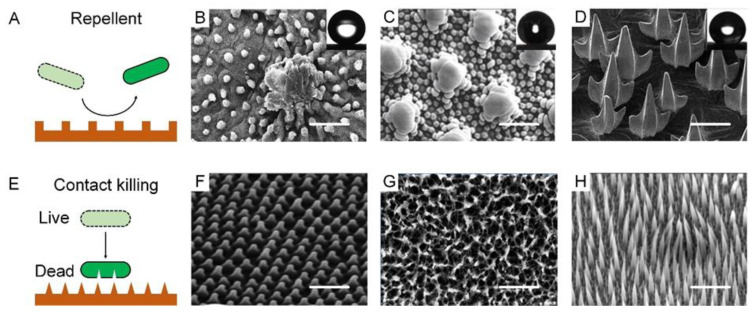
Nature bacteria-repellent surface and contact-killing surfaces. (**A**) Schematic diagram of natural bacteria-repellent surface. (**B**) Lotus leaves. SEM images showing “micropapilla” structures in lotus leaves. Scale bar: 50 µm. Reproduced with permission [54]. 2008, American Institute of Physics (United States). (**C**) Springtail skin. SEM images showing “doubly-reentrant” structures in springtail skin. Scale bar: 2 µm. Reproduced with permission [34]. 2017, Wiley-VCH. (**D**) Sharkskin. SEM images showing “microdenticle” structures in sharkskin. Scale bar: 100 µm. Reproduced with permission [55]. 2019, Nature Portfolio. (**E**) Schematic diagram of natural contact-killing surface. (**F**) Cicada wings. SEM images showing “nanocone” structures in cicada wings. Scale bar: 2 µm. Reproduced with permission [56]. 2012, Public Library of Science. (**G**) Dragonfly wings. SEM images showing “disordered nanopillar” structures in dragonfly wings. Scale bar: 200 nm. Reproduced with permission [49]. 2013, Nature Portfolio. (**H**) Lizard skin. SEM images showing “microspine” structures in lizard skin. Scale bar: 2.5 µm. Reproduced with permission [51]. 2017, Nature Portfolio.

## 4. Bioinspired Antibacterial Surfaces

In-depth studies of natural antibacterial surfaces offer viable strategies for developing high-performance bioinspired counterparts to effectively prevent bacterial infection [23,24]. A summary of current antibacterial surfaces is shown in Table 1 [57,58,59]. Briefly, depending on whether extra interventions are needed, bioinspired antibacterial surfaces can be categorized into passive and active ones (Figure 3).

### 4.1. Passive Antibacterial Surface

#### 4.1.1. Bacteria-Repellent Surface

Boosting the surface repulsion of bacteria, which is mainly inspired by superhydrophobic biological skins, can remarkably minimize the bacterial infection rate. Generally, the factors that can control bacterial repellence include wettability, topography, material stiffness, surface charge, and their combinations.

The most common approach to interfere with the surface–bacteria interaction is to regulate the surface wettability, i.e., hydrophobicity or hydrophilicity (Figure 4(Ai)). For example, the surface can be rendered hydrophobic by grafting low-surface-energy molecules or infusing liquid lubricant, the result of the latter being named slippery liquid-infused porous surfaces (SLIPSs) [60,61,62]. A novel SLIPS consisting of microporous poly (butyl methacrylate-co-ethylene dimethacrylate) films infused with the perfluoropolyether fluid−slippery poly (butyl methacrylate-co-ethylene dimethacrylate) was demonstrated to prevent different strains of the opportunistic pathogen *P. aeruginosa* from biofilm formation for 7 days [63]. Further combining low-surface-energy components with a large microroughness can amplify the apparent wettability, which tremendously weakened bacterial adhesion [64,65]. One typical example was a lotus-leaf-inspired surface patterned with regularly spaced micro-pillar arrays and packed nanoneedles, allowing a more than 98% antibacterial rate against *Escherichia coli* at high-concentration (10^8^ colony-forming unit/mL) and long-term-culture conditions [66,67,68,69,70,71]. A mimic of shark skin with superhydrophobicity also showed a similar inhibition to the adhesion of the zoospores Ulva (~5 μm diameter) and *S. aureus* (1 μm diameter) [72,73,74]. 

On the other hand, causing surface hydrophilicity can also decrease the total contact and inhibit bacterial adhesion, because the hydrophilicity helps to reduce the number of bacteria proteins attached to the surface. A simple way to make surfaces hydrophilic is directly coating hydrophilic components such as polymers and zwitterions [75,76,77,78,79]. For example, a branched-chain-polymer-based surface with antibiofouling properties was developed by conjugating dioxy-containing polyethylene glycol with gentamicin terminals. The introduction of polyethylene glycol increased the surface hydrophilicity, which inhibited protein adhesion and repelled bacterial fouling. In addition, the transplanted *S. aureus* infection model showed that the branched-chain polymers have good antibacterial and antifouling ability in vivo. (Figure 4(Aii)) [80]. For zwitterions, the introduction of zwitterions onto cotton-texture surfaces significantly increased surface hydrophilicity. The modified cotton texture surfaces can effectively resist initial bacterial adhesion, kill attached bacteria, and release dead bacteria [81,82,83]. 

Solely topographical modifications also provide a persistent and predictable form for control of bacterial behavior, especially using ordered patterns (Figure 4(Bi)). M. Yang et al. showed that submicron-scale pillar patterns strikingly inhibit bacterial adhesion, growth, and colonization by physically hindering bacterial cell-to-cell interactions. Furthermore, they investigated the effect of morphology (e.g., honeycomb) and sizes on the adhesion and growth of bacterium with different shapes (e.g., rod *E. coli* and spherical *S. aureus*). The fluorescent image results showed that a 1-μm patterned surface significantly reduced bacterial adhesion and growth while inhibiting bacterial colonization compared with a flat surface (Figure 4(Bii)). From a dynamic perspective, the selective adhesion of bacterial cells to patterns was synergistically mediated by maximizing cell–substrate contact area and minimizing cell deformation. They established that two main factors, namely energetically favorable adhesion sites and physical confinement, contribute to the antibacterial properties of the honeycomb-like pattern [84].

Adopting soft materials, e.g., hydrogel, with low stiffness, can tune the surface bacterial adhesion (Figure 4(Ci)). Harder polymer surfaces typically have higher network densities than softer polymer surfaces, resulting in a higher density of functional groups that liquid media and bacterial cells can interact with. Generally, a soft hydrogel surface with low stiffness exhibits better antibacterial performance. A positive correlation between the surface stiffness and adhesion was demonstrated by larger bacteria colonization on the stiffer surface (Figure 4(Cii)). Polyelectrolyte multilayer membranes from polyacrylamine hydrochloride and polyacrylic acid were prepared with Young’s moduli ranging from 1 to 100 MPa. A positive correlation between the surface stiffness and adhesion with *E. coli* and *S. aureus* was found on such surfaces [85]. A cross-linked membrane composed of poly(L-lysine) and hyaluronic acid was prepared, and the number of bacteria on the non-cross-linked membrane at 30 kPa was lower than that on the cross-linked membrane at 150 kPa [86]. Polydimethylsiloxane substrates with a stiffness of 100 to 2600 kPa were found that affected the physiology of *E. coli* and *P. aeruginosa* [87]. A positive relationship between the fouling intensity of *E. coli* and *S. aureus* and hydrogel stiffness was reported by conducting tests of bacterial attachment on three poly (ethylene glycol) dimethacrylate surfaces with low (44.05–308.5 kPa), moderate (1495–2877 kPa), and high (5152–6489 kPa) stiffness, respectively [88].

The additional benefit of the polymeric modification is its static charge, which can interact with bacterial membranes (Figure 4(Di)). In general, most bacterial cells are surrounded by a layer of peptidoglycan (composed of sugars and amino acids) that are negatively charged, which can be trapped or even killed on positively charged polymeric surfaces, or repelled by negatively charged surfaces. However, this repellency is largely dependent on the species of bacteria. For example, Gram-positive bacteria with a polycationic glycocalyx were more likely to adhere to negatively charged surfaces than Gram-negative bacteria with a polyanionic glycocalyx. Due to the larger discharge capacity, the direct current positive charging method had a better antibacterial effect than the alternating current charging method. In addition, the capacitance-based platform can effectively prevent the formation of biofilms by means of cyclic charging. Extracellular electron transfer between bacteria and charged titania nanotubes doped with carbon-disrupted bacterial morphology and induced intrabacterial ROS burst, leading to bacterial death upon charging (Figure 4(Dii)) [89].

**Figure 4 biomimetics-07-00088-f004:**
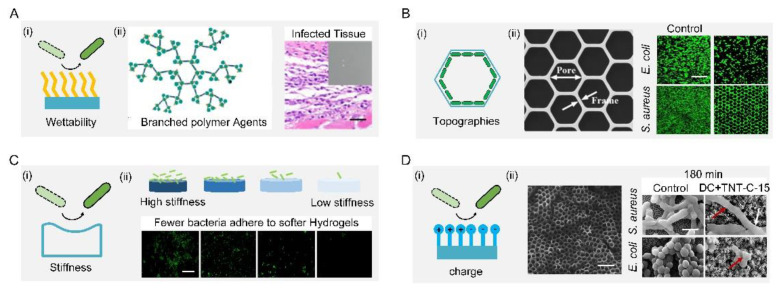
Bacteria-repellent surfaces. (**A**) Surface hydrophilization by modifying antibiofouling polymer (**i**) antibiofouling polymeric agents and their application for the surface functionalization of Ti substrate; the resultant surface can effectively repel bacterial adhesion in vivo (**ii**). Scale bar: 100 µm. Reproduced with permission [80]. 2018, American Chemical Society (United States). (**B**) Schematic diagram of topographies’ surfaces (**i**) SEM and fluorescent images of selective adhesion of bacterial cells on the honeycomb platform (**ii**). Scale bar: 50 µm. Reproduced with permission [84]. 2015, Elsevier. (**C**) Schematic diagram of stiffness surface (**i**) schematic diagram of the antifouling property of stiffness surface (**ii**). Scale bar: 10 µm. Reproduced with permission [88]. 2015, American Chemical Society (United States). (**D**) Schematic diagram of charge surface (**i**) Scale bar: 500 nm. Schematic diagram of the antifouling property of charge surface (**ii**) Scale bar: 2 μm. Reproduced with permission [89]. 2018, Nature Portfolio.

#### 4.1.2. Contact-Killing Surfaces

It has to be admitted that bacteria-repellent surfaces cannot always successfully prevent bacteria from attaching to them. In this case, we need another effective strategy to resist bacterial infection, namely contact-killing surfaces, where bacteria are killed once they come into contact with the surface. Contact-killing surfaces can be designed and engineered via coating with bactericidal layers or tuning mechanical properties.

Bactericidal substances such as antibacterial metal, antibacterial polymers, and antibacterial peptides can be covalently immobilized on the surfaces [90]. Antibacterial metals should be toxic to a broad spectrum of bacteria, such as Zn^2+^, Na^+^, Mg^2+^, Ca^2+^, K^+^, Ag^+^, Hg^2+^, and As^3+^, most of which have been used as antibacterial agents since ancient times [91]. While how antibacterial metals kill bacteria has not yet been fully understood, two possible mechanisms have been proposed to try to explain this phenomenon. First, antibacterial metal could generate oxidative stress to form reactive oxygen species that can kill bacteria. For example, Au was added to the Pd catalyst to promote the release of oxygen-based radical species. It was found that this method was more bactericidal and virucidal and inhibited biofilm formation compared to other methods based on chlorination or pre-formed H_2_O_2_ alone [92,93]. Second, the release of free metal ions from metal surfaces was responsible for bacteria inactivation (Figure 5(Ai)) [94,95,96,97,98,99,100,101,102,103,104]. For example, compared with a pure strontium calcium phosphate coating, the addition of Zn^2+^ increased the killing rates for *S. aureus* and *E. coli* from 61.25% and 55.38% to 83.01% and 71.28%, respectively. Bacteria on such surfaces with Zn^2+^ underwent partial shrinking, twisting, and even dissolving before death (red arrows) (Figure 5(Aii)) [105].

Negative charges enrich surfaces and generate an attraction force that destroys the integrity of cell membranes and inactivates bacterial enzymes. Most negatively charged materials are polymers, which can be covalently bonded with surfaces with long-term effectiveness. Commonly used antibacterial polymers include quaternary ammonium compounds (QACs), quaternary phosphoniums, and N-chloramines. Taking QACs as an example, they have strong contact-killing activity against both Gram-positive bacteria and Gram-negative bacteria by destroying their membrane (Figure 5(Bi)). A QAC (s-poly (2,3-dimethylmaleic anhydride) (melittin)-b-poly (2-hydroxyethyl methacrylate) was modified on a surface as a multistage polymer brush to combat bacterial infection [106]. However, these QAC-based surfaces tended to induce irritation and inflammation, which hindered their practical application in the biomedical field [107,108,109,110,111,112,113,114]. In contrast to antibacterial polymers, antibacterial peptides hold great potential to solve the issues of irritation and inflammation and reduce the possibility of induced resistance. When interacting with negatively charged bacterial membranes, antibacterial peptides that usually carry a positive charge in the physiological environment would self-assemble into secondary structures such as α-helical structures, β-sheet structures, ring structures, extended structures, and mixed structures. These shape changes of antibacterial surfaces induced by self-assembly exposed their initially hidden amino acid, which can destroy the integrity of a cell membrane and further kill bacteria [115,116,117]. For example, WRWRWR-G_4_-(dihydroxyphenylalanine)_4_ was allowed to first self-assemble, and then screws with this coating were implanted into femurs near the joints of Sprague–Dawley rats to evaluate their antibacterial performance in vivo. This animal experiment indicated that the number of bacteria on both the screws and the surrounding tissues were reduced compared with those on bare screws (Figure 5(Bii)), indicating good antibacterial properties [118].

**Figure 5 biomimetics-07-00088-f005:**
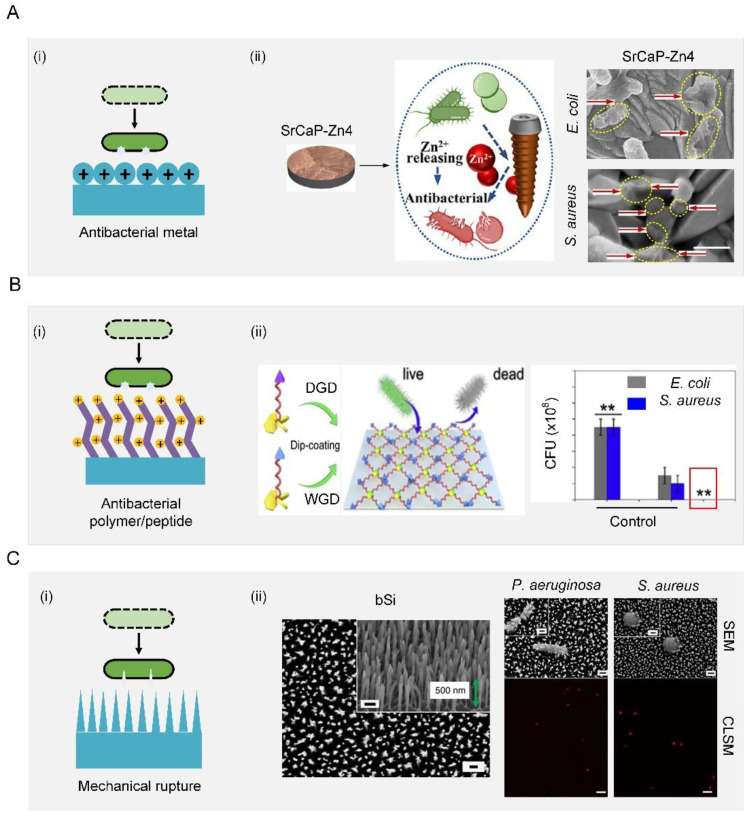
Contact-killing surfaces. (**A**) Schematic diagram of an antibacterial metal surface (**i**) Schematic illustration of antibacterial metal for synergistic photothermal/pharmacological antibacterial therapy (**ii**). Scale bar: 500 nm. Reproduced with permission [105]. 2021, Elsevier. (**B**) Schematic diagram of antibacterial polymer/peptide surface (**i**) schematic illustration of antibacterial polypeptide WRWRWR-G_4_-(dihydroxyphenylalanine)_4_ and their application for the surface functionalization of medical implants (**ii**). Reproduced with permission [118]. 2019, Elsevier. (**C**) Schematic diagram of mechanical rupture surface (**i**) SEM image of the upper surface of black silicon. Scale bars: 200 nm. SEM images of *P. aeruginosa* and *S. aureus* were significantly disrupted through interaction with black silicon. Scale bars, 200 nm. CLSM images confirmed that disruption by black silicon was lethal to the cells; non-viable bacterial cells and spores were stained with propidium iodide (red) whereas the living cells were stained with SYTO 9 (green). Scale bars: 5 mm (**ii**). Reproduced with permission [50]. 2013, Nature Portfolio.

The contact-killing function can also be achieved by the mechanical rupturing of cells using fine surface structures, such as nanopatterns [119-121], nanowires [122], nanotubes [123], and nanopillars [124,125,126,127,128,129,130,131,132,133,134,135,136]. The rupturing of the bacterial membrane occurs when the cell membrane is not elastic enough to bear the exerted tensile force (Figure 5(Ci)). For example, a dragonfly-wing-inspired surface patterned with nanopillars possessed a mechanical bactericidal effect. As a result, this surface was highly bactericidal against all Gram-negative and Gram-positive bacteria. It showed an estimated average kill rate of up to 450,000 cells/min cm^2^. The cell integrity of the bacteria was mechanically disrupted by the patterned nanopillars on the surface. Moreover, the viability analysis of bacteria using confocal laser scanning microscopy (CLSM) confirmed that all bacteria were dead after attachment (red color) (Figure 5(Cii)). This biomimetic work demonstrated promising prospects for the development of a new generation of antibacterial surfaces [50].

#### 4.1.3. Responsive Surfaces

The aforementioned passive antibacterial surface shares a bottleneck: the killing efficiency of antibacterial surfaces may become weakened while the toxicity to normal cells and tissues may be exacerbated during long-term use. A potential alternative candidate is responsive surfaces that can be switched on/off in an on-demand manner [137]. Surface responsiveness is attributed to the use of agents that can be stimuli-triggered by the change in certain bacterial chemical cues (i.e., pH and enzymes) or external triggers (i.e., temperature, ions, light, and magnetism).

Bacterial infections are always accompanied by acidification of the environment (the pH of the infection site drops to 5.5). Such a change in pH can act as a powerful trigger to turn on the antibacterial function by exposing a surface-bound bactericide or releasing preloaded antibacterial agent [138]. The acidic effect mainly stems from low-oxygen fermentation triggering the production of organic acids, such as lactic acid secreted by *S. aureus* or acetic acid secreted by *E. coli*. (Figure 6(Ai)). Such characteristics can be harnessed to selectively release the antibacterial substances, killing bacteria in real time. For example, a hierarchical antibacterial surface was constructed with a top layer of pH-responsive polymer brush and a bottom layer of bactericidal agents. Decreases in pH could collapse the top layer and induce the exposure of the bactericidal agents, and ultimately activate the bactericidal function. More importantly, the recovery of pH could reconfigure the top layer and switch off the bactericidal function, demonstrating reversibility [139]. When the pH drops from 7.4 to 5.0, the killing efficiency of the proposed surface changes from 9.3% to 77.5% (Figure 6(Aii)). 

In addition to pH, substances secreted by bacteria during metabolism, such as enzymes, can also act as a powerful trigger for killing activity (Figure 6(Bi)). For example, an enzyme-responsive peptide biointerface was designed based on the saliva-acquired pellicle bioinspired polypeptide DDDEEKRWRWRWGPLGVRGD (SAP-MP196-G-1) that consists of the enzyme-responsive sequence GPLDV and the antimicrobial peptide RWRWRW. When the biointerface was invaded by *S. aureus*, the enzyme response sequence GPLDV was cleaved by the secreted enzyme from *S. aureus*. As a result, the antimicrobial peptide RWRWRW was exposed to kill bacteria. By measuring the number of bacteria in the different groups through quantification of OD_600_, it was found that bacterial growth was markedly lower on the proposed surface than in a control. (Figure 6(Bii)) [140].

An ion-responsive surface can be achieved by grafting specific ion-pair polymers on the surface, which can endow surfaces with conformational change, high surface wettability, and electrostatic repulsion under the action of external ions. The additional ion-responsive polymer consists of anionic and cationic ionizable units in each repeating unit. The strong hydration of anionic and cationic ionizable units makes the surface excellent in antibiofouling properties. With such beneficial characteristics, the ion-responsive surface appeared a good candidate for achieving antibiofouling function through ion variation (Figure 6(Ci)) [141,142]. As a three-function surface, the reusable antibacterial surface was prepared with comprehensive antibiofouling, antibacterial, and self-cleaning properties. This antibacterial surface comprised (1) poly-n-hydroxyethyl acrylamide hydrophilic polymer as an ultralow-pollution background that can prevent long-term bacterial colonization; (2) triclosan, which can effectively kill attached bacteria; (3) a salt-sensitive polymer, namely, poly(3-(dimethyl(4-vinylbenzyl) amino) propyl sulfonate), which was used to release attached bacteria in the salt solution. The antibacterial surface exhibited three functional antimicrobial activities: poly-n-hydroxyethyl acrylamide resisted bacterial attachment, triclosan killed about 90% of the bacteria on the surface, and poly(3-(dimethyl(4-vinylbenzyl) amino) propyl sulfonate) released about 90% of the dead bacteria on the surface (Figure 6(Cii)) [143].

The temperature has been widely used to control antimicrobials on solid surfaces made of thermally responsive polymers. One typical example of thermally responsive polymers is poly(n-isopropylacrylamide) (PNIPAAm), which can be utilized to achieve a temperature-responsive surface with wettability for bacterial adhesion and separation (Figure 6(Di)) [144,145]. When the temperature rises higher than the lower critical solution temperature of PNIPAAm, the hydrogen bond between PNIPAAm and water is severely broken, resulting in the hydrophobicity of PNIPAAm. A new thermally responsive surface consisting of thermally responsive hydrogel regions and mechanically supported elastomer regions were prepared. The alternative microscale arrangement of these two regions enabled the surface morphology to have a significant effect on disrupting bacterial colonization and dispersing heat-sensitive individual bacteria. This can effectively prevent bacterial infection without inducing the cohesive loss of human epidermal tissue, thereby serving as an extracellular biointerface for precise local antimicrobial therapy (Figure 6(Dii)) [146].

Many surfaces are sensitive to light, including ultraviolet, visible, or near infra-red light. In practical applications, visible or near infra-red light is more attractive for clinical applications due to its low toxicity and deep tissue penetration. For example, light-responsive surfaces with antibacterial strategies such as antimicrobial photothermal therapy (APTT) and antimicrobial photodynamic therapy (APDT) rely on the generation of local antimicrobial properties to kill cells driven by different frequencies of light (Figure 6(Ei)). APTT is a physical antibacterial strategy, in which the photothermal agent can continuously heat up under specific light, and the high temperature induces cell-membrane rupture, protein/enzyme denaturation, cell cavitation, and cell-fluid evaporation [147]. For example, two-dimensional Nb_2_C Mxene nanosheets as a photothermal agent with implanted medical titanium plates were prepared. The temperature of modified titanium plates was raised steadily to 70 °C within 2 min under the irradiation of a high-power density near-infrared laser. The bacterial survival rates for *S. aureus* and *E. coli* dropped sharply from 100.4% ± 3.12% and 100.02% ± 2.76% in the control group to 1.19% ± 0.93% and 1.06% ± 0.58% in the modified titanium plates + near-infrared laser group, respectively [148]. APDT is a minimally invasive strategy that uses light-responsive photosensitizers to generate reactive oxygen species through photochemical reactions, resulting in irreversible damage and cell death [149,150]. For example, smart nanoplatforms with photosensitizer molecule chlorin e6 were prepared. When light irradiated the above platform, the ratio of anaerobic *P. gingivalis* and *Fusobacterium nucleatum* was reduced from 66.21% in the control group to 51.91% [151]. Combing APTT and APDT, a red phosphorus/zinc oxide heterojunction was prepared that has excellent solar photothermal conversion and photocatalytic efficiency, further leading to the death of bacteria through hyperthermia and reactive oxygen species. The bacteriostatic effectiveness on *S. aureus* at 5 min was 99.96 ± 0.03%, and that on *E. coli* at 4 min was 99.97 ± 0.02% (Figure 6(Eii)) [152].

A magnetic responsive surface is also a good antibacterial surface. The application of the magnetic field induced the magnetic metal to spin, deform, and exert physical forces on the bacteria, which resulted in the disruption of the dense biofilm matrix and simultaneous lysis of the cells. Once exposed to a low-intensity rotating magnetic field, the liquid metal droplets are physically driven to change shape, creating sharp edges. When in contact with bacterial biofilms, the particle motion created by the magnetic field, coupled with the presence of nanoedges, physically ruptures the bacterial cells and disrupts the dense biofilm matrix. For example, magnetic galinstanc-based liquid metal platforms can also kill bacteria under an external magnetic field (Figure 6(Fi)). After introducing two major pathogens biofilms, specifically *P. aeruginosa* and *S. aureus*, the system was exposed to a dynamic magnetic field of 775 mGs. Following 90 min of exposure to the magnetic field with the gallium-based liquid metal ferrofluid platforms, it was observed that the average colony-forming unit /mm^2^ was reduced for both biofilms of *S. aureus* (99.85%, *p* 0.001) and *P. aeruginosa* (96.51%, *p* 0.01) when compared to controls (Figure 6(Fii)) [153].

**Figure 6 biomimetics-07-00088-f006:**
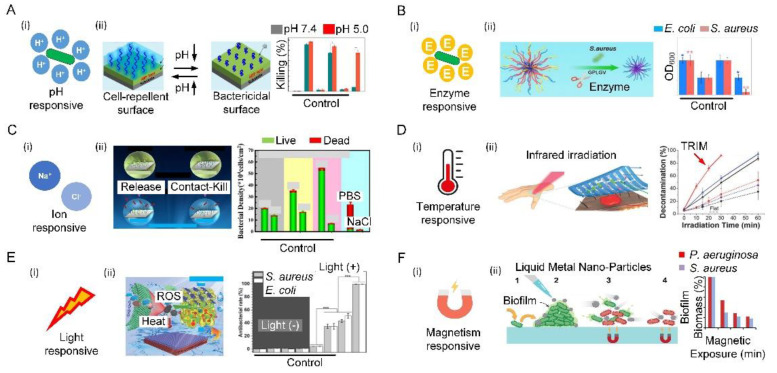
Bioinspired responsive surfaces. (**A**) Schematic illustration of pH-responsive antibacterial surfaces (**i**) a hierarchical surface, when bacteria colonize surfaces, bacteria-induced acidification collapses the outermost poly (methacrylic acid) chains, thereby exposing the underlying antimicrobial peptides to kill bacteria on demand. Additionally, dead bacteria can be released after the PMAA chains recover hydrophilicity due to an increase in ambient pH (**ii**). Reproduced with permission [141]. 2016, American Chemical Society (United States). (**B**) Schematic diagram of the enzyme-responsive antibacterial surface (**i**) illustration of the bio-interface with enzyme-responsive, antibacterial, and cell-adhesion functions for tissue engineering (**ii**). Reproduced with permission [142]. 2021, Elsevier. (**C**) Schematic diagram of ion-responsive antibacterial surface (**i**) salt-responsive poly(3-(dimethyl(4-vinylbenzyl) ammonio) propyl sulfonate), antifouling poly (N-hydroxyethyl acrylamide), and bactericidal triclosan to form two types of brushes, which demonstrated their tri-functional antibacterial activity to resist bacterial attachment by poly (N-hydroxyethyl acrylamide), to release ~90% of dead bacteria from the surface by poly(3-(dimethyl(4-vinylbenzyl) ammonio) propyl sulfonate), and to kill ~90% of bacteria on the surface by triclosan (**ii**). Reproduced with permission [145]. 2019, Royal Society of Chemistry. (**D**) Schematic illustration of temperature-responsive antibacterial surfaces (**i**) the thermal-disrupting interface induced mitigation film attached to the skin, absorbing infrared light and generating localized heat that kills bacteria in the recessed areas of the film while sparing surrounding epithelial host cells (**ii**). Reproduced with permission [146]. 2020, Wiley-VCH. (**E**) Schematic illustration of light-responsive antibacterial surfaces (**i**) Schematic illustration of photocatalytic and photothermal efficiency of red phosphorus/zinc oxide heterojunction thin film for rapid point-of-use disinfection (**ii**). Reproduced with permission [152]. 2019, Wiley-VCH. (**F**) Schematic illustration of magnetism-responsive antibacterial surfaces (**i**) Magnetically responsive gallium-based liquid metal droplets act as antimicrobial materials to physically damage, decompose, and kill pathogens within mature biofilms (**ii**). Reproduced with permission [153]. 2020, American Chemical Society (United States).

### 4.2. Active Antibacterial Surface

Unlike the biological antibacterial surfaces that are adaptive and flexible to diverse harsh environments, an artificial surface usually displays relatively short-term bacterial resistance. To prevent bacterial adhesion and biofilm formation, new physical removal strategies relying on external sources such as mechanical force or energy waves have emerged as alternatives to an antibacterial agent. Generally, these external sources include shear force, interfacial tension, mechanical waves, dynamic actuating motions, and plasma treatment. Unlike passive antibacterial surfaces, the antibacterial process of the active antibacterial surface is controllable and acts directly on bacteria.

The shear-force-based method is very effective to remove bacteria directly by generating shear force sufficient to balance the bacterial adhesion force (Figure 7(Ai)). Shear forces can be produced by the external application of force parallel to the surface. The inner wall of the microfluidic device is composed of the copolymers, 2-methacryloyloxyethyl phosphorylcholine, 3-methacryloxypropyl trimethoxysilane and 3-(methacryloyloxy) propyl-tris(trimethylsilyloxy) silane, with two typical thicknesses (20 and 40 nm), forming a cross-linked film. Shear forces were generated by friction between the fluid and the inner wall of the microfluidic device. Under the same shear stress, thicker surfaces could weaken the adhesion of S. aureus, which leads to more bacteria detachment (Figure 7(Aii)) [154].

Surface antibacterial performance can also be improved by adjusting the interfacial tension with surfactants. Among those, one effective way to control interfacial tension is the use of biosurfactant, a kind of surface-active biomolecule produced by many microorganisms. It has been reported that the aggregation of biosurfactants at the interface can reduce the interfacial tension of the solution and form a microcellular structure, which can disrupt the bacterial cell membrane to produce antibacterial properties (Figure 7(Bi)). Moreover, the properties of the biosurfactant itself, such as its concentration, also influence the antimicrobial performance. Taking sophorolipids (a type of biosurfactant) as an example, at a concentration above 5% *v*/*v*, they can inhibit the growth of Gram-negative *Cupriavidus necator* and Gram-positive *Bacillus* sp. with a bactericidal effect. Below this concentration, the antibacterial properties are greatly reduced [155]. In addition, Zein/gum Arabic nanoparticles were prepared to stabilize the oil–water interface of Pickering emulsions, which strikingly inhibited the growth of *E. coli*. The stabilized emulsion exhibited a controlled release and the antibacterial activity of thymol due to the protective effect from its stable interfacial layer (Figure 7(Bii)) [156].

Mechanical waves such as ultrasound waves could induce surfaces made of piezoelectric materials to generate reactive oxygen species to make them antibacterial (Figure 7(Ci)). For example, a piezoelectric surface was prepared by using barium titanate nanocubes whose Schottky junctions were modified with gold nanoparticles. This surface could sense exogenous ultrasound waves and produce highly reactive oxygen species as a response to obtain antibacterial ability (Figure 7(Cii)). It was demonstrated that this surface exhibited high antibacterial efficiency against both typical Gram-negative and Gram-positive bacteria, offering a promising method for efficient ultrasonic therapy [157]. 

Dynamic actuating motions of surfaces can prevent bacterial attachment to suppress surface fouling (Figure 7(Di)). This phenomenon has been widely found in nature, such as red blood cells, arteries, blood vessels, starfish, seaweed, mussels, and the skin of batoidea and pilot whales. Particularly, batoidea manipulate their body in an undulatory style to generate vortices to repel bacteria. Inspired by this, a flexible multilayer responsive surface was designed to integrate dynamic undulatory motion with bactericidal nanospine arrays (Figure 7(Dii)). Under an applied magnetic field, this surface behaved with a batoidea-like undulatory motion, which generated strong vortices to repel bacteria. Moreover, the integration of a dynamic undulatory motion and static nanospine array enabled this surface to repel and kill bacteria simultaneously, effectively inhibiting biofilm formation for an extended period of 7 days [158].

Plasma treatment is commonly used to make surfaces antibacterial owing to its ability to change the surface wettability. For example, after being treated by non-thermal atmospheric pressure plasma jets (NTAPPJs), the titanium surface became antibacterial against two bacteria with different cell-wall structures, including Gram-positive and Gram-negative bacteria. The adhesion and biofilm formation rates of bacteria on NTAPPJ-treated titanium surfaces were significantly reduced compared to untreated samples. Surfaces treated with NTAPPJ can induce oxidation in bacteria, which were more susceptible to Gram-negative bacteria due to differences in the cell-wall structure. In samples treated with NTAPPJs for a longer time, the adhesion rate and biofilm formation rate of Gram-negative bacteria were significantly lower than those of Gram-positive bacteria [159].

**Figure 7 biomimetics-07-00088-f007:**
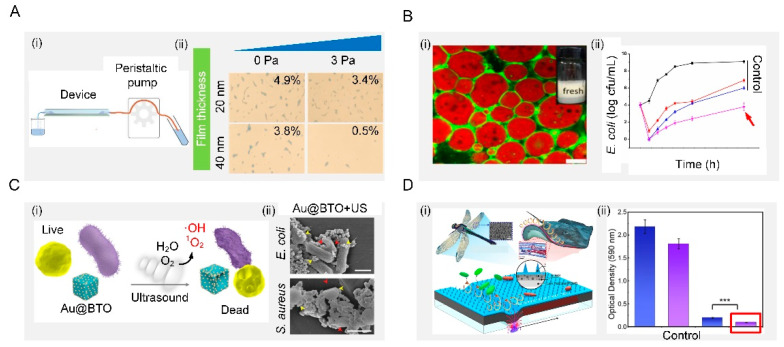
Active removal. (**A**) Shear removal. Schematic illustration of the process of bacterial adhesion strength assay in the microfluidic device. After pre-seed of *S. aureus* in a microfluidic device, many bacterial cells detached from the C-40 surface under the same shear stress. Reproduced with permission [154]. 2016, Royal Society of Chemistry. (**B**) Interfacial tension. Schematic illustration of fresh Pickering emulsions. Effects of Zein/GA-thymol Pickering emulsions on the growth of *E. coli.* Reproduced with permission [156]. 2021, Elsevier. (**C**) Mechanical waves. Schematic diagram of sonodynamic bacterial elimination based on piezoelectric nanocomposite. Reproduced with permission [157]. 2021, Elsevier. (**D**) Dynamic actuating motions. Conceptual schematic of dynamic nanoneedle composites and long-term antibiofilm testing. Reproduced with permission [158]. 2021, Elsevier.

## 5. Applications of Bioinspired Antibacterial Surfaces

### 5.1. Biomedical Devices/Implants

The past 30 years have seen the explosive development of biomedical devices and implants, such as hips, knee-implant screws, stents, heart valves, and blood-vessel grafts, for saving lives and tremendously restoring the quality of human life. However, these devices/implants face the risk of function failure induced by bacterial infection, which may lead to secondary damage to patients or even death. Therefore, the quest for durable biomedical equipment to control bacterial attachment and multiplication has become ever more urgent [160,161,162,163,164]. In pursuit of this, S. Sang et al. used polydopamine as intermediate and loaded strontium carbonate nanoparticles on porous sulfonated PEEK materials through inlays to obtain strontium-doped SPEEK, which could promote bone formation and osseointegration ability. Subsequently, they loaded a layer of silk protein-gentamicin coating on the SPEEK material to endow it with antibacterial ability. This provided innovative inspiration for developing novel functional orthopedic implants, such as antibacterial and antiloose implants [165].

### 5.2. Wound Dressing

Open wounds and burns are susceptible to bacterial infection due to the lack of protection of the superficial skin. A traditional dressing was made from silver or an antibacterial agent, or was manufactured with slow-release silver or an antibacterial agent while considering safety issues such as the induction potential and the consumption of biological substances from this slow-spreading material [166]. For example, the preparation of durable antibacterial wound dressings by using the strong interfacial interaction between polyhydroxy antibiotics and gelatin and its in situ cross-linking with polydopamine [167]. However, an antibiotic-based dressing can easily induce multidrug resistance in bacteria and even the emergence of super bacteria, greatly limiting its practical application. As an alternative, researchers have explored bioinspired dressings by mimicking the way the host immune system detects and eliminates bacteria. A bioinspired dressing with a catechol-chitosan film-like melanin structure and redox activity was prepared. The film had reversible redox activity and could catalyze the repeated transfer of electrons from ascorbic acid to oxygen to continuously generate reactive oxygen species. Furthermore, this bioinspired dressing can also generate reactive oxygen species, impart antibacterial activity and promote wound healing [168]. Considering the uncertainty and possibility of pathogen invasion during a relatively long wound-healing process, an ideal dressing should be able to monitor wound conditions and effectively suppress pathogens in a timely way. Toward this end, a multifunctional wound dressing based on a novel self-healing elastomer was developed, which can enable real-time monitoring of temperature, pH, and glucose in the healing area and sutureless closure (Figure 8A) [169].

### 5.3. Electronic Skin

Electronic skin with functions like natural skin has attracted intense interest in a variety of applications including wearables, person-centric health monitoring, smart prosthetics and robotics, and human-machine interfaces. Since electronic skin is usually attached to the human body, it is prone to microbial growth due to the contamination from sweat containing organic matter. Such a microbial growth could lead to inflammation and bacterial infections in the users, which are detrimental to their health. To fundamentally overcome this issue, electronic skin should be endowed with additional antibacterial ability on top of its original function. On one hand, the antibacterial properties can remove the interference of bacteria on the electronic skin and obtain more stable bioelectronic signals, including electrocardiography and electromyography information (Figure 8B). At the same time, the electronic skin had a strong inherent antibacterial effect on *E. coli* and *S. aureus*, effectively limiting the development of microorganisms and preventing bacterial infection, and thus providing users with a comfortable and safe environment [170]. On the other hand, a self-driven antibacterial electronic skin with antibacterial was built based on an all-nanofiber triboelectric nanogenerator. This electronic skin enabled real-time, self-powered monitoring of whole-body physiological signals and joint movements. More importantly, the electronic skin had excellent antibacterial activity against *E. coli* and *S. aureus*, which can greatly inhibit bacterial growth and prevent bacterial infection [171].

### 5.4. Air Disinfection System

The disinfection of air, water, equipment surfaces and even the human body is indispensable for disease prevention and public safety, playing a crucial role in the health of human beings and the sustainable development of society. As population and human activities increase rapidly, the uncontrolled discharge of polluted air and wastewater has been irreversibly damaging human health, natural resources, and biological chains. For example, such pollution would produce a large number of airborne pathogens, the carrier of many bacteria that could cause the transmission of pneumonia, asthma, and influenza. In response to these great threats to public health, urgent action was therefore needed to address such pollution at the source. One typical example was a self-powered disinfection system that can quickly disinfect airborne bacteria and viruses (Figure 8C). This system relied on an efficient nanowire-assisted electroporation mechanism, which was actuated by a vibration-driven triboelectric nanogenerator. More than 99.99% of bacteria and viruses are inactivated in the air at a fast airflow rate (2 m/s), which is equivalent to a processing time of 0.025 s while maintaining a low pressure drop of only 24 Pa. This work provides a proof-of-concept demonstration for the practical application of green ventilation systems in buildings [172].

**Figure 8 biomimetics-07-00088-f008:**
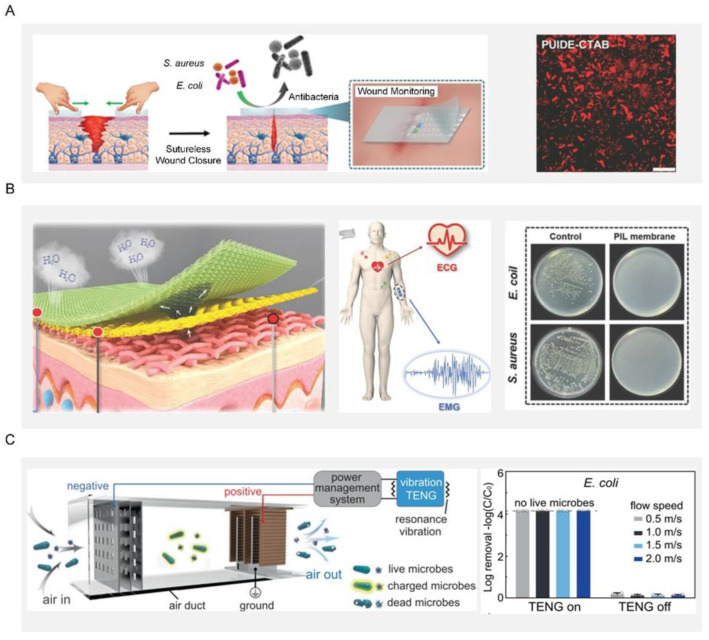
Applications of bioinspired antibacterial surfaces. (**A**) Schematic diagram of self-healing elastomer multifunctional wound dressing. Bacteria after being cocultured with PUIDE-CTAB for 6 h. Reproduced with permission [169]. 2021, Wiley-VCH. (**B**) Schematic diagram of fabrication and application of PIL-membrane electronic skin with hygroscopic, breathable, and antibacterial properties for bioelectric signal monitoring. Reproduced with permission [171]. 2022, Wiley-VCH. (**C**) Schematic diagram of the resonance-vibration-driven disinfection system in the air duct. Disinfection efficiency for *E. coli*. Reproduced with permission [172]. 2021, Nature Portfolio.

## 6. Conclusions and Perspectives

Bacteria have and will coexist with humans forever; as such, how do humans survive an environment with ubiquitous bacteria that remarkably impact our health and ecological safety? Considering that conventional solutions such as antibacterial agents are very likely to lose their effectiveness when used frequently, developing alternative methods that are smart, eco-friendly, and effective long-term has become more and more urgent. Fortunately, the emergence of bioinspired antibacterial surfaces that mimic optimal natural strategies has shown enormous potential in preventing bacterial infection. In this review, we first described the fundamental process of bacterial infection and the factors that may potentially affect the process. We next summarized several typical natural antibacterial surfaces and their antibacterial mechanisms. We also presented the key bioinspired surfaces that can break down the bacterial transmission chain. The excellent performance of these bioinspired surfaces demonstrates the feasibility of them serving as potential alternatives to antibacterial agents. Despite their remarkable capability to suppress bacterial invasion, bioinspired antibacterial surfaces are facing some limitations and concerns that hinder their practical applications [173].

To promote the transition from lab research to practical applications, the overall performance of the bioinspired antibacterial surfaces should be improved enough to cope with complex, transient scenarios in the real world [76,174,175,176,177,178,179,180,181,182,183,184]. First, a bioinspired antibacterial surface should be highly adaptive to a wide spectrum of bacteria. Currently, a certain bioinspired antibacterial surface is designed to be solely effective against a certain type of bacteria. Such a one-to-one relationship is ineffective in a practical environment teeming with a variety of bacteria and needs to be upgraded to a one-to-many relationship, in which a given surface can defend against infections from multiple types of bacteria. Toward this end, the surface may be fundamentally redesigned from the perspectives of topology and materials to integrate various abilities effective against different bacteria. Briefly, a multiscale hierarchical structure is realized by combining bacteria-repellent surfaces, such as lotus-leaf, springtail-skin, and shark-skin structures, with contact-killing surfaces such as cicada-wing, dragonfly-wing, and lizard-skin structures. This multiscale hierarchical structure can achieve the repelling or contact-killing of a broad spectrum of bacteria, even fungi and viruses. Implementing the two functions simultaneously can require resorting to decomposing them into multiscale hierarchical structures. Second, a bioinspired antibacterial surface should be highly durable for long-term use. At present, those surfaces suffer from function deterioration owing to the long-time exposure to the working environment, greatly impairing the antibacterial performance. By contrast, such a problem is not found in their natural counterparts living in moist environments. Therefore, an innovative bottom-up strategy is required to fundamentally reveal natural mechanisms and design bioinspired surfaces that can rival their natural counterparts in terms of functionality and durability. Third, bioinspired antibacterial surfaces need to achieve biocompatibility. For now, bioinspired antibacterial surfaces remain controversial due to the biological complexity of medical conditions. A low-biocompatibility surface can lead to blood clotting and all manner of other inflammation and tissue-sensitization problems. The above problems can be improved by optimizing biocompatible materials and bionics design. Bioinspired antibacterial surfaces with biocompatibility can reduce inflammation and increase the success rate of implants.

As for the transformation of bioinspired antibacterial science to actual productivity, the fabrication technique for bioinspired antibacterial surfaces needs to be revolutionized to realize some essential advantages including their large-scale, cost-effective, and eco-friendly use. First, large-scale fabrication capability, while not sacrificing the quality and performance of the surfaces, is fundamental to driving mass adoption, which is, however, inaccessible in current strategies that commonly rely on microfabricated equipment. Specifically, the manufacture of high-precision lithography machines for surface processing is a difficult problem. Second, manufacturers are constantly trying to find ways to make their processes more cost-effective, including continuously developing advanced and sophisticated production machinery, improving cutting tools, and optimizing overall cutting systems. Specific strategies include high-speed machining, high-feed machining, high-performance machining, and digital machining [173]. Third, eco-friendliness is also very important in the processing process. In the production of materials, or as raw materials, or due to technological requirements, many processes inevitably introduce some harmful substances, which not only cause pollution in production and worsen labor conditions but also cause long-term damage to human health and the environment after they are transformed into products.

Focusing on the biomedical applications of the bioinspired antibacterial surfaces, safety becomes the first and foremost priority. Currently, it is still controversial whether the organic or inorganic compounds contained in bioinspired antibacterial surfaces will cause adverse effects on mammalian cells during long-term use. In this vein, the toxicity of bioinspired antibacterial surfaces resulting from extra-immune responses should be minimized or even eliminated before they enter the market. Indeed, achieving full commercialization of bioinspired antibacterial surfaces involves highly interdisciplinary problems, ranging from chemistry and engineering to biomedicine and clinical medicine, and still has a long way to go. To bridge this gap, one feasible solution may be to decompose the highly interdisciplinary problems into smaller and more manageable pieces. This calls for continuous and collective efforts from chemists, engineers, biomedical researchers, and clinicians to design, develop, demonstrate, and deliver next-generation bioinspired antibacterial surfaces and products. We envision that, with the support of constant effort, remarkable breakthroughs will be made in the development of bioinspired antibacterial surfaces, which will further enrich the library of antibacterial strategies for human health.

## Figures and Tables

**Figure 1 biomimetics-07-00088-f001:**
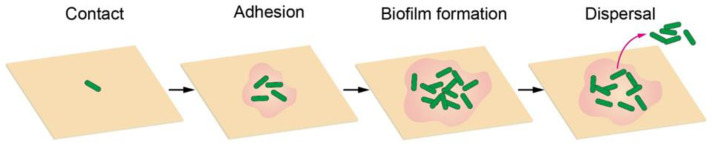
Schematic diagram of bacterial adhesion and biofilm formation process.

**Figure 3 biomimetics-07-00088-f003:**
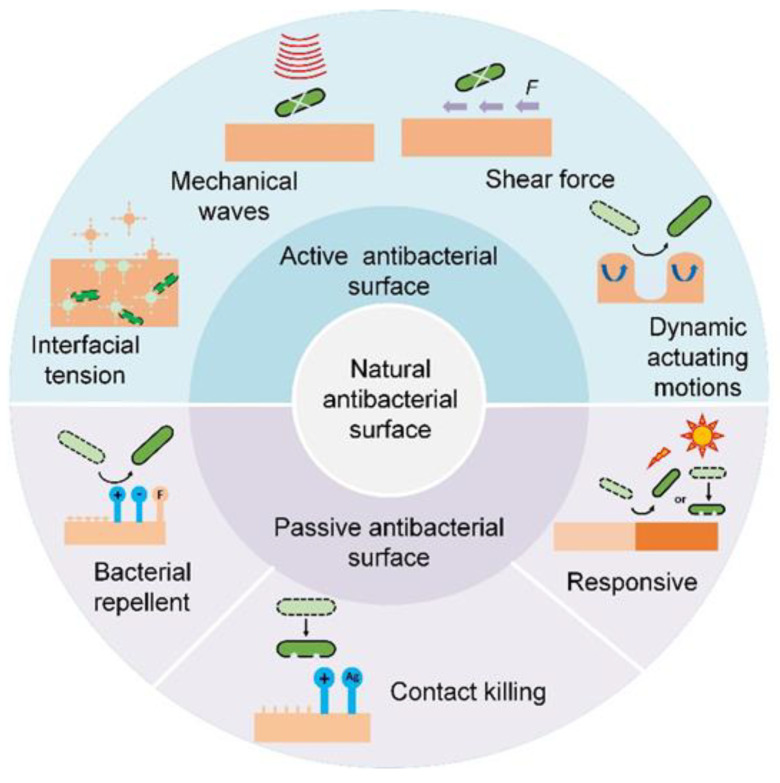
Scheme of nature and bioinspired antibacterial surfaces.

**Table 1 biomimetics-07-00088-t001:** Current surface-modification approaches in the design of passive bioinspired antibacterial surfaces.

Approach	Preparation Methods	Antibacterial Effects	Comments
Bacteria-Repellent Sur-face	Contact-Killing Surface
Passive bioinspired antibacterial surface	Chemical modification	Surface-initiated polymerization	Antibiofouling polymer	Antibiotics	Uneven
2.Vapor-deposition polymerization	2.Fluoride	2.Antibacterial metal	2.Mechanically weak
3.Electrospinning	3.Zwitterion	3.Antibacterial polymer	3.Lack long-term stability
4.Sedimentation	4.SLIPSs	4.Antibacterial peptide	4.Concentration dependent and needs further chemical reactions
5.Spin-coating			
6.Lay-er-by-layer deposition			
7.Oil-impregnation			
Physical modification	Plasma etching	Black silicon	Fast processing
2.Hydrothermal etching	2.Ti nanowires	2.High precision
3.Anodic oxidation	3.Titanium dioxide nanotubes	3.Needs the assistance of related processing equipment to complete
4.Magnetron sputtering	4.Nanocolumnar thin film on Si substrate	
5.Epitaxial growth	5.2D honeycomb structure	
6.Exfoliation	6.Single-layer graphene sheet	
7.Chemical vapor deposition	7.Carbon nanotube “forest”	
8.Nano-imprint lithography	8.Thin needle-like structure	
9.Templating	9.Homogenous monolayered films/graphene flakes	
10.Electrospinning	10.Sharp high-aspect-ratio structures	
	11.Slightly rounded	
	12.High aspect ratio structures	
	13.Superhydrophobic fibrous mat	
Bioinspired responsive surface		1.pH-responsive	
	2.Enzyme-responsive	
	3.Temperature-responsive	
	4.Ion-responsive	
	5.Light-responsive

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
