# Peer review of "Recent Progress on Bioinspired Antibacterial Surfaces for Biomedical Application"

_biomimetics, 2022, doi:10.3390/biomimetics7030088_

Round 1

Reviewer 1 Report

Recent Progress on Bio-inspired Antibacterial Surfaces for Biomedical Application

This paper overall is of great structure and carries interesting information. See specific comments below:

Some of the figures need improvements, specifically: Figure 4 bcd very hard to see the smaller inserts of fluorescent or EM images. Scale bars and labels are also too small in front. Figure 6 all lables and words on the bar graph, line graph axis labels, figure c (ii) all fronts are way too small. 

The idea of following the categorization of repellent vs killing is very good. 

Figure 4: part A (ii) this is actually unclear, needs more explanation in the text about what the branched polymer is uniquely enabling, and what is the interpretation of the histology slide with the gray box. C. stiffness is a very complicated idea and relevant to scale - things can be very stiff on nano-scale but still makes a soft overall matrix, 

Figure 5: A - the red arrows are not clear, please indicate what geometry or shape that is focused on. 

Figure 6: the red box are not helpful. Also part B (ii) is confusing. 

Author Response

Reviewer 1 

This paper overall is of great structure and carries interesting information. See specific comments below:

  1. Some of the figures need improvements, specifically: Figure 4 bcd very hard to see the smaller inserts of fluorescent or SEM images. Scale bars and labels are also too small in front. Figure 6 all lables and words on the bar graph, line graph axis labels, figure c (ii) all fronts are way too small. 

Response: Thanks for your comments. We have enlarged inserts, scale bars and labels in Figure 4 and all labels and words on the bar graph, line graph axis labels, figure c (ii) in Figure 6. The new Figure 4 and Figure 6 were attached.

Figure 4

Figure 6

  1. The idea of following the categorization of repellent vs killing is very good. 

Response: Thanks for your comments.

  1. Figure 4: part A (ii) this is actually unclear, needs more explanation in the text about what the branched polymer is uniquely enabling, and what is the interpretation of the histology slide with the gray box. C. stiffness is a very complicated idea and relevant to scale - things can be very stiff on nano-scale but still makes a soft overall matrix, 

Response: Thanks for your comments. In addition, the transplanted S. aureus infection model showed that the branched-chain polymers have good antibacterial and antifouling ability in vivo. Harder polymer surfaces typically have higher network densities than softer polymer surfaces, resulting in a higher density of functional groups that liquid media and bacterial cells can interact with.

  1. Figure 5: A - the red arrows are not clear, please indicate what geometry or shape that is focused on. 

Response: Thanks for your comments. We used yellow dotted line to draw the geometry or shape. The new Figure 5 was attached.

Figure 5

  1. Figure 6: the red box are not helpful. Also part B (ii) is confusing. 

Response: Thanks for your comments. We have removed the red box and redraw the part B (ii). The new Figure 6 was attached.

Figure 6

Reviewer 2 Report

This review manuscript presents a brief summary of the latest development and progress on the bio-inspired antibacterial surfaces, the author presented the recent progress in bioinspired active and passive antibacterial strategies. The biomedical applications and the prospects of these antibacterial surfaces were fully discussed. This manuscript is generally well organized, I recommend its publication after a revision, and some comments are given as follows:

1.Pg2.  L79-89, the authors mentioned that flagella also play an important role in adhesion. Whether the structure of the flagella and the protein transport process involved in the adhesion process can be explained in detail?

2. Pg7, Line239, the authors mentioned that adopting soft materials can tune the surface bacterial adhesion, the reason for this needs to be explained in detail.

3. Pg11. Line 450, A detailed description of how the magnetic field leads to the disruption of the dense biofilm matrix and simultaneous lysis of the cells is suggested.

Author Response

Reviewer 2

This review manuscript presents a brief summary of the latest development and progress on the bio-inspired antibacterial surfaces, the author presented the recent progress in bioinspired active and passive antibacterial strategies. The biomedical applications and the prospects of these antibacterial surfaces were fully discussed. This manuscript is generally well organized, I recommend its publication after a revision, and some comments are given as follows:

  1. L79-89, the authors mentioned that flagella also play an important role in adhesion. Whether the structure of the flagella and the protein transport process involved in the adhesion process can be explained in detail?

Response: Thanks for your comments. As bacterium comes into contact with a surface, their appendages, such as flagella, will inevitably interact with it. The interaction between flagella and surfaces could enhance adhesion, because of the inherent hydrophobicity of flagella which allows to adhere to hydrophobic surfaces. By contrast, the presence of flagella may also weaken adhesion as found in Caulobacter crescentus. Therefore, the influence of flagella on adhesion is much more complex and fully understanding it requires in-depth investigations. In addition to the flagella, some other filamentous protein extensions on the cell surface, including fimbriae, curli and pili, are also involved in nonspecific initial adhesion to abiotic surfaces. For example, pili can use their specific receptors to bind to substrates through an unidentified mechanism, and most pili show no preference to substrates.

  1. Pg7, Line239, the authors mentioned that adopting soft materials can tune the surface bacterial adhesion, the reason for this needs to be explained in detail.

Response: Thanks for your comments. Harder polymer surfaces typically have higher network densities than softer polymer surfaces, resulting in a higher density of functional groups that liquid media and bacterial cells can interact with.

  1. Line 450, A detailed description of how the magnetic field leads to the disruption of the dense biofilm matrix and simultaneous lysis of the cells is suggested.

Response: Thanks for your comments. Once exposed to a low-intensity rotating magnetic field, the liquid metal droplets are physically driven to change shape, creating sharp edges. When in contact with bacterial biofilms, the particle motion created by the magnetic field, coupled with the presence of nanoedges, physically ruptures the bacterial cells and disrupts the dense biofilm matrix.

Reviewer 3 Report

Review of biomimetics-1790203

This is a very nice and comprehensive review of bio-inspired antibacterial surfaces. There are several issues that need to be addressed to improve this manuscript even better, as follows:

  1. Line 727: What do you mean with “new power”? Please elaborate.
  2. For all scientific names, please write the genus in complete way (not abbreviated) at the first time the name of an organism is mentioned. At the second time, the genus can be abbreviated. The scientific names:

·       Line 37: Staphylococcus aureus and Pseudomonas aeruginosa

·       Line 211: Escherichia coli

·       Line 162: Porphyromonas gingivalis

·       Line 443: Fusobacterium nucleatum

  1. Please check again the name of ALL of the authors in the references. Please do not remove the middle name. For example:
    Reference 41: Chien, H.-W., Chen, X.-Y., Tsai, W.-P., Lee, M.

Reference 62: …Busscher, H.J., van der Mei, H.C. … 

  1. Line 10: Cardiovascular -->NOT “carAdioVASUlar”
  2. Line 47-48: Please kindly add this additional article to broaden the knowledge of silver not only for the application of wound disinfection and drinking water disinfection, but also as a catalyst for organic waste degradation: Food & Bioproducts Processing 121 (2020) 193-201 https://doi.org/10.1016/j.fbp.2020.02.008
  3. Line 105: Perhaps please change “interdicting” with a simpler or more common word.
  4. Line 109-111: Please delete this portion, which is from the original MDPI template.
  5. Section 3.1: Please kindly add this additional article of shark skin-inspired structure for preventing biofilm formation: Materials Letters 285 (2021) 129098  https://doi.org/10.1016/j.matlet.2020.129098

  1. Line 151: Gram-negative…Gram-positive --> please use uppercase G, since it is from the name of Hans Christian Gram, a prominent Danish bacteriologist.
  2. Line 152: Bacillus sp. --> please add sp. (in italic)
  3. Line 199-225: Please split this long paragraph.
  4. Line 245: Please write scientific names in italic.
  5. Line 259:… Yet, this repellency is largely…
  6. Line 285: It has to be admitted that…
  7. Line 516: …(a type of biosurfactant)…
  8. Line 517-518:…Gram-negative…Gram-positive --> please use uppercase G
  9. Line 518: Bacillus sp. --> please add sp. (in italic)
  10. Line 708: …ecofriendliness is also…
  11. Reference 25: Vibrio cholera --> Please write scientific names in italic.
  12. Reference 47: Escherichia coli --> Please write scientific names in italic.
  13. Reference 69: Oryza sativa--> Please write scientific names in italic.
  14. Reference 71: Escherichia coli --> Please write scientific names in italic.
  15. Reference 82: Colloid Surfaces B --> uppercase S
  16. Reference 88: ACS Nano --> uppercase N
  17. Reference 96: Pseudomonas aeruginosa --> Please write scientific names in italic.
  18. Reference 134: Ceriops tagal --> Please write scientific names in italic.
  19. Reference 158: TiO2 --> subscripted 2

Author Response

Reviewer 3

This is a very nice and comprehensive review of bio-inspired antibacterial surfaces. There are several issues that need to be addressed to improve this manuscript even better, as follows:

  1. Line 727: What do you mean with “new power”? Please elaborate.

Response: Thanks for your comments. To make it clear, we changed ‘new power’ to ‘the constant efforts’ which refers to the continuous efforts from chemists, engineers, biomedical researchers, and clinicians as aforementioned in our manuscript. As a result, the sentence now is ‘We envision that, under the support of the constant efforts, remarkable breakthroughs will be made in the development of bioinspired antibacterial surfaces, which will further enrich the library of antibacterial strategies for human health.

  1. For all scientific names, please write the genus in complete way (not abbreviated) at the first time the name of an organism is mentioned. At the second time, the genus can be abbreviated. The scientific names:
  • Line 37: Staphylococcus aureusand Pseudomonas aeruginosa
  • Line 211: Escherichia coli
  • Line 162: Porphyromonas gingivalis
  • Line 443: Fusobacterium nucleatum

Response: Thanks for your comments. The name of the genus has been changed.

  1. Please check again the name of ALL of the authors in the references. Please do not remove the middle name. For example:
    Reference 41: Chien, H.-W., Chen, X.-Y., Tsai, W.-P., Lee, M.

Reference 62: …Busscher, H.J., van der Mei, H.C. … 

Response: Thanks for your comments. The middle name of the authors has been added.

  1. Line 10: Cardiovascular -->NOT “carAdioVASUlar”

Response: Thanks for your comments. The mistake has been corrected.

  1. Line 47-48: Please kindly add this additional article to broaden the knowledge of silver not only for the application of wound disinfection and drinking water disinfection, but also as a catalyst for organic waste

degradation: Food & Bioproducts Processing 121 (2020) 193-201 https://doi.org/10.1016/j.fbp.2020.02.008

Response: Thanks for your comments. The reference has been added.

  1. Line 105: Perhaps please change “interdicting” with a simpler or more common word.

Response: Thanks for your comments. We replaced ‘interdicting’ with ‘preventing’ to make it more readable.

  1. Line 109-111: Please delete this portion, which is from the original MDPI template.

Response: Thanks for your comments. The portion has been deleted.

  1. Section 3.1: Please kindly add this additional article of shark skin-inspired structure for preventing biofilm formation: Materials Letters285 (2021) 129098  https://doi.org/10.1016/j.matlet.2020.129098

Response: Thanks for your comments. The reference has been added.

  1. Line 151: Gram-negative…Gram-positive --> please use uppercase G, since it is from the name of Hans Christian Gram, a prominent Danish bacteriologist.

Response: Thanks for your comments. The mistake has been corrected.

  1. Line 152: Bacillus sp.--> please add sp. (in italic)

Response: Thanks for your comments. The mistake has been corrected.

  1. Line 199-225: Please split this long paragraph.

Response: Thanks for your comments. The mistake has been corrected.

  1. Line 245: Please write scientific names in italic.

Response: Thanks for your comments. The mistake has been corrected.

  1. Line 259:… Yet, this repellency is largely…

Response: Thanks for your comments. The mistake has been corrected.

  1. Line 285: It has to be admitted that…

Response: Thanks for your comments. The mistake has been corrected.

  1. Line 516: …(a type of biosurfactant)…

Response: Thanks for your comments. The mistake has been corrected.

  1. Line 517-518:…Gram-negative…Gram-positive --> please use uppercase G

Response: Thanks for your comments. The mistake has been corrected.

  1. Line 518: Bacillus --> please add sp. (in italic)

Response: Thanks for your comments. The mistake has been corrected.

  1. Line 708: …ecofriendliness is also…

Response: Thanks for your comments. The mistake has been corrected.

  1. Reference 25: Vibrio cholera--> Please write scientific names in italic.

Response: Thanks for your comments. The mistake has been corrected.

  1. Reference 47: Escherichia coli--> Please write scientific names in italic.

Response: Thanks for your comments. The mistake has been corrected.

  1. Reference 69: Oryza sativa--> Please write scientific names in italic.

Response: Thanks for your comments. The mistake has been corrected.

  1. Reference 71: Escherichia coli--> Please write scientific names in italic.

Response: Thanks for your comments. The mistake has been corrected.

  1. Reference 82: Colloid Surfaces B --> uppercase S

Response: Thanks for your comments. The mistake has been corrected.

  1. Reference 88: ACS Nano --> uppercase N

Response: Thanks for your comments. The mistake has been corrected.

  1. Reference 96: Pseudomonas aeruginosa--> Please write scientific names in italic.

Response: Thanks for your comments. The mistake has been corrected.

  1. Reference 134: Ceriops tagal--> Please write scientific names in italic.

Response: Thanks for your comments. The mistake has been corrected.

  1. Reference 158: TiO2--> subscripted 2

Response: Thanks for your comments. The mistake has been corrected.
